

# Meta-SourceTracker: application of Bayesian source tracking to shotgun metagenomics

Jordan J. McGhee[1], Nick Rawson[2], Barbara A. Bailey[2],
Antonio Fernandez-Guerra[3,5], Laura Sisk-Hackworth[4] and Scott T. Kelley[4]

[1] Bioinformatics and Medical Informatics Program, San Diego State University, San Diego, CA, United States of America
[2] Department of Mathematics and Statistics, San Diego State University, San Diego, CA, United States of America
[3] Microbial Genomics and Bioinformatics Research Group, Max Planck Institute for Marine Microbiology, Bremen, Germany
[4] Department of Biology, San Diego State University, San Diego, CA, United States of America
[5] Current affiliation: Lundbeck Foundation GeoGenetics Centre, GLOBE Institute, University of Copenhagen, Copenhagen, Denmark

Corresponding author
Scott T. Kelley, skelley@sdsu.edu, skelley@mail.sdsu.edu

## ABSTRACT

**Background**. Microbial source tracking methods are used to determine the origin of contaminating bacteria and other microorganisms, particularly in contaminated water systems. The Bayesian SourceTracker approach uses deep-sequencing marker gene libraries (16S ribosomal RNA) to determine the proportional contributions of bacteria from many potential source environments to a given sink environment simultaneously. Since its development, SourceTracker has been applied to an extensive diversity of studies, from beach contamination to human behavior.

**Methods**. Here, we demonstrate a novel application of SourceTracker to work with metagenomic datasets and tested this approach using sink samples from a study of coastal marine environments. Source environment metagenomes were obtained from metagenomics studies of gut, freshwater, marine, sand and soil environments. As part of this effort, we implemented features for determining the stability of source proportion estimates, including precision visualizations for performance optimization, and performed domain-specific source-tracking analyses (i.e., Bacteria, Archaea, Eukaryota and viruses). We also applied SourceTracker to metagenomic libraries generated from samples collected from the International Space Station (ISS).

**Results**. SourceTracker proved highly effective at predicting the composition of known sources using shotgun metagenomic libraries. In addition, we showed that different taxonomic domains sometimes presented highly divergent pictures of environmental source origins for both the coastal marine and ISS samples. These findings indicated that applying SourceTracker to separate domains may provide a deeper understanding of the microbial origins of complex, mixed-source environments, and further suggested that certain domains may be preferable for tracking specific sources of contamination.

## INTRODUCTION

Microbes are found in every environment, from the depths of the Pacific Ocean to the hostile conditions of the Atacama Desert. Most microbes co-exist with other microbes in biofilms or in complex dynamic communities referred to as 'microbiomes' (e.g., the gut microbiome) that include hundreds or thousands of different microbial species, many of which play critical roles in animal health and ecosystem function. While much is known about the species composition of microbial communities, less is understood about how they form in the first place and how microbes move among different ecosystems. Understanding the origins of microbial communities is particularly important for tracking routes of contamination, such as polluted water systems, but also has important implications for understanding microbiome development and ecosystem function.

Microbial source tracking (MST) approaches have been developed to determine the source origins of particular microbes, with their primary use being the study of bacterial contamination of municipal water (*Liu et al., 2018*), natural freshwater systems (streams, rivers and lakes) (*Newton et al., 2013*; *Staley et al., 2018*), and coastal ocean waters. Standard MST approaches track microbial sources via one or more key bacterial strains or species previously linked to a specific source (e.g., *E. coli* strains only found in cow feces). Traditional MST methods rely on techniques such as culture isolation and PCR with species-specific primers. Other MST approaches have relied on patterns of multiple antibiotic-resistance and carbon utilization profiles (*Simpson, Santo Domingo & Reasoner, 2002*; *Scott et al., 2002*). More recently, improvements in next-generation sequencing (NGS) technologies has resulted in NGS being widely in all aspects of microbiology including MST (*Van Dijk et al., 2014*; *Martin et al., 2018*).

The widely-used SourceTracker program has provided one of the most powerful and effective methods for using NGS data to perform MST (*Knights et al., 2011*). This program uses a combination of Bayes' theorem and Gibbs sampling to analyze data from large bacterial 16S rRNA marker-gene NGS libraries. Unlike previous MST methods, which use individual microbes to identify routes of colonization and contamination, SourceTracker uses data from hundreds or thousands of species, and allows simultaneous estimation of the proportion of multiple source environments contributing to a given sink environment, including an estimate of unknown sources. SourceTracker uses Bayesian methods to evaluate all assignments of sink sequences to all source samples, including an unknown source, and creates a joint distribution of those assignments. Then, the distribution is sampled with a Gibbs sampler to estimate the likelihood that a sequence came from a particular source. For example, in a study of bacterial assemblages on restroom surfaces, the researchers used SourceTracker to estimate the relative proportion of skin, feces and soil contributing to each specific sink sample (*Flores et al., 2011*). At the time of this writing, SourceTracker had been cited over 400 times with a surprising diversity of applications, including identifying individuals within the same species based on their microbiomes, determining which body sites contribute most to contamination of built environments and detecting sources of early gut colonization (*Flores et al., 2011*; *Hewitt et al., 2013*; *Hyde et al., 2016*; *Chen et al., 2018b*; *Kapono et al., 2018*). Other applications included applying

SourceTracker to forensic analysis and studies of human behavior (*Lax et al., 2015*; *Bik et al., 2016*). SourceTracker was designed for use with bacterial 16S rRNA marker genes and has primarily been used with these data. However, it has been applied to a few shotgun metagenomic studies, including one that tracked the source origins of antibiotic resistance gene markers (*Baral et al., 2018*). While more expensive to generate and more computationally intensive to analyze, shotgun metagenomic data provide a much broader potential array of microbial diversity (Bacteria, Archaea, Eukaryota, and viruses) for use in microbial source tracking.

In this study, we tested a metagenomic-SourceTracker approach, a novel application of the SourceTracker software for metagenomic data, to determine the source origins of complex microbial samples. Our goals were twofold: First, to test the effectiveness of SourceTracker for metagenomic data with samples of known origins, and secondly to provide tools for determining the reliability of proportion estimates. (We refer to the processing of metagenomic data and application of SourceTracker to metagenome data as metagenomic-SourceTracker, or mSourceTracker for short.) We tested mSourceTracker with metagenome samples collected from coastal marine environments, which are commonly a mix of different sources due to runoff from freshwater environments and contamination from land debris. We also used mSourceTracker to determine the sources of contamination in samples collected from the International Space Station (ISS). Our results showed that mSourceTracker provides a robust approach for microbial source tracking with metagenomic datasets and further demonstrated how mSourceTracker can be used to provide domain-specific biological insights into the movements of microbes among ecosystems.

## MATERIALS & METHODS

### Data collection

Metagenome sequence libraries were obtained from samples collected from coastal marine water, fresh water, human gut (feces), sand and soil environments from multiple studies. These environments were chosen as likely sources of microorganisms to be found in coastal marine waters, which tend to have runoff from rivers and possibly contaminated with sewage. Source samples are selected from environments likely to contribute organisms found in the sink samples. For example, metagenome samples from human skin would be appropriate source samples for an mSourceTracker analysis of samples from surfaces frequently touched by human hands (e.g., computer keyboards). A total of 223 metagenome samples were used for this study: 110 coastal marine samples, 30 freshwater samples, 64 soil samples, 6 sand samples, and 13 gut samples (see Table S1 for details and accession numbers from the European Nucleotide Archive).

Sequence libraries from metagenomic studies of the International Space Station (ISS), human skin, human gut, and soil were downloaded from ENA for a second mSourceTracker analysis. Soil, human skin, and human gut samples were selected as probable sources of microbes in the ISS microbiome. 82 metagenome samples were used for this analysis: 24 samples each from soil, human skin, human gut, and 10 ISS samples (see Table S2 for details and accession numbers.)

## Taxonomic separation of metagenomic data

All metagenomic samples were preprocessed with fastp (version 0.2) to remove adaptors and low-quality sequence reads (*Chen et al., 2018a*). Taxonomic abundances were generated for all samples using the k-mer approach implemented in the Kaiju ver. 1.5.0 program (*Menzel, Ng & Krogh, 2016*). Kaiju produces estimated taxonomic abundances primarily at the genus level. To determine the domain of each genus (Archaea, Bacteria, Eukaryota or virus), we wrote a programming script in python3.6 using the URL:

http://taxonomy.jgi-psf.org/tax/sc_name/{} to extract the full taxonomic lineage information given the genus name.

For example, given the genus Salmonella, the URL: http://taxonomy.jgi-psf.org/tax/sc_name/Salmonella returns the string:

sk:Bacteria;p:Proteobacteria;c:Gammaproteobacteria;o:Enterobacterales;f:Enterobacteriaceae;g:Salmonella

The "genus" names within the Kaiju output that did not return a lineage from the URL were manually searched in NCBI. The domain information was then added to a dictionary within our script and later compiled into a single data frame. For domain specific source tracking analysis, the taxonomic abundances for each sample were separated by domain and written into corresponding data frames. Species and count numbers from each sample were merged using species name with all previously processed samples within each domain-specific data frame. Once all samples were processed through our pipeline, the assembled data frames for each domain were then written to an output table formatted to be used with mSourceTracker. Example data, code, and instructions to perform this step may be found in the Kaiju_Table_to_OTU_table folder at: https://github.com/residentjordan/SourceTracker2-diagnostics.

Kaiju version 1.7.2 generated taxonomic abundances for samples in the ISS analysis dataset. This version of Kaiju returns the entire taxon string. For each sample, read count numbers were merged into one data frame using the taxon string to make the metagenome data frame. The metagenome data frame was then divided by domain, then each domain and metagenome data frames were written to separate output files. Counts for species with incomplete taxa strings were removed from the files. These files were converted to the HDF5 biom format using the biom-format package.

## Simulation data for mSourceTracker analysis

To test the prediction accuracy of mSourceTracker, 14 samples from the coastal marine environment were defined as sinks. The rest were of the coastal marine samples were used as sources, a set which did not include the 14 sink samples. The analysis was performed on the combined dataset and each domain separately, with a default rarefaction limit of 1,000. Samples that did not have a minimum count of 1,000 for any given kingdom file were removed from all datasets. Proportions for sink samples were compared using 10 and 100 draws. The number of chains was set at 5 for all comparisons. The same mapping file was used for comparing differences in proportions between kingdom datasets. The number of chains was held at 5 and the number of draws were variable so as to keep the chain differences below 5%.

As 100 draws and 5 chains were established as sufficient for chain convergence, those values were used to calculate source proportions for the 10 ISS sink samples from gut, soil, and skin sources. An mSourceTracker analysis with 5 chains (100 draws/chain) needed between one and two hours to run on MacOS with a 3.4 GHz processor and 32 GB of RAM.

## mSourceTracker: diagnostic add-on feature

We wrote a new function for SourceTracker2, called via the '–diagnostics' option. This new option allows user to determine how well the Markov chains are converging in any given sample. This option was not available with the original SourceTracker or in the SourceTracker2 original repository. It can be accessed along with scripts, sample test data, an instruction manual, and supplementary analysis results at: https://github.com/residentjordan/SourceTracker2-diagnostics. In SourceTracker 2, Gibbs sampling data computed from the SourceTracker2 'envcounts' array are written to a temporary output file along with source and sink ID's. When the diagnostic function is called using the command '–diagnostics', the Gibbs data file is read and placed into an array using numpy (*Van der Walt, Colbert & Varoquaux, 2011*). The array is split based on the number of chains and number of draws defined by user inputs. Here, we use the standard Markov Chain Monte Carlo terminology of "draw" and "chain". However, it should be noted that the original SourceTracker2 codebase uses the term "restart" to refer to an MCMC draw, and "draw" for an MCMC chain. Array data is multiplied by the alpha1 preset to convert numbers into respective proportion values. Each chain produces a moving average via Gibbs sampling over the number of draws selected. The script then calculates the difference between the maximum and minimum chains. If the proportion value of any two chains differs by a default value of 5%, or by user defined parameters, all chains are exported onto a single line graph per sample for each environment. Each line represents a single chain and the legend displays the proportion estimate of each chain for the given sample and environment. A single text output table displays the absolute differences between the maximum and minimum chains for all samples in each environment.

## Random forest classifier

Sample names in the feature tables used for mSourceTracker analysis were converted to the name of the environment from which they were collected. One source was then selected to be tested with all other sources being categorized as 'other' to identify features important in classifying the selected environment. Training and testing sets were randomly created at approximately a 3:1 ratio. Random Forest is an ensemble learning method which classifies by the votes of its component trees. Using the scikit-learn Random Forest classifier (*Pedregosa et al., 2011*) we fit and classified the data using 500 trees. Random state was set at 0 and the out-of-bag score was made True. The classifier was run multiple times to ensure there were no important features returned due to overfitting or other errors. The 10 most important features were graphed for each environment based on relative importance utilizing pandas (*McKinney, 2010*) and matplotlib (*Hunter, 2007*). Confusion matrices and statistics for the Random Forest classifier were also produced using scikit-learn modules.

This process was performed for each of the organismal domains. A template script, sample dataset, and instructions for performing the random forest and confusion matrix analysis are provided in the SourceTracker2-diagnotics GitHub folder "Random_Forest."

## RESULTS

### Effects of parameter adjustments on the accuracy and precision of mSourceTracker

In order to apply the Bayesian approach of mSourceTracker to metagenomics data, we downloaded a total of 223 samples from 5 clearly identified environments. These sources include 110 coastal marine samples, 30 freshwater samples, 64 soil samples, 6 sand samples, and 13 gut samples. Fourteen of the coastal marine samples were chosen as "sink" samples (indicated in boldface on Table S1). These 14 sink samples were excluded from the coastal marine source sample set. The k-mer based Kaiju analysis identified a total of 5,725 taxa across all sample from the four major taxonomic domains of life. Of these, 88.8% of them were bacterial sequences. Eukaryotes and archaea comprised ∼9.0% and ∼1.9% of our metagenomic sequences, respectively. Viral sequences comprised just ∼0.3% of all the samples.

Since previous studies indicated that adjusting SourceTracker's default parameters (e.g., number of restarts) with 16S data led to more stable estimates of source proportions (*Henry et al., 2016*), we determined how the proportional composition for our sink samples would be affected if we adjusted the default parameters for metagenomic samples. Figures 1A and 1B show how the number of draws affected proportion estimates. Because the estimates are a moving average, increasing the number of draws to 100 resulted in a decreased variability between each chain. We also increased the number of chains to 5 so we could compare multiple independent proportional estimates in a single run. As indicated in Fig. 1C, with only 10 draws source proportion estimates among the different chains could vary considerably but increasing the number of draws to 100 (Fig. 1D) resulted in convergence of the chains. Analysis of 14 source proportion estimates from 223 samples using 5 chains with 10 draws per chain found an average of $3 \pm 2\%$ difference between the two most different estimates.

Increasing the number of restarts to 100 substantially minimized the differences among chains. Figures 1E–1H) demonstrate how increasing the number of draws and chains reduced the variability in the source proportion estimates for metagenomic samples. As mentioned previously, each draw is dependent on prior draws and a single chain runs the risk of getting caught in a local maximum in the target distribution and returning an inaccurate estimation. More draws reduce the likelihood this phenomenon could affect the final estimations because draws are averaged together for each environment. For all subsequent testing we adjusted the default parameters in mSourceTracker to minimize the range between chains such that the biggest difference between chains would be less than 5%. Increasing the number of draws to 200 did not significantly reduce the variability among chains (see SourceTracker2-diagnostics GitHub folder "Chain_Convergence").

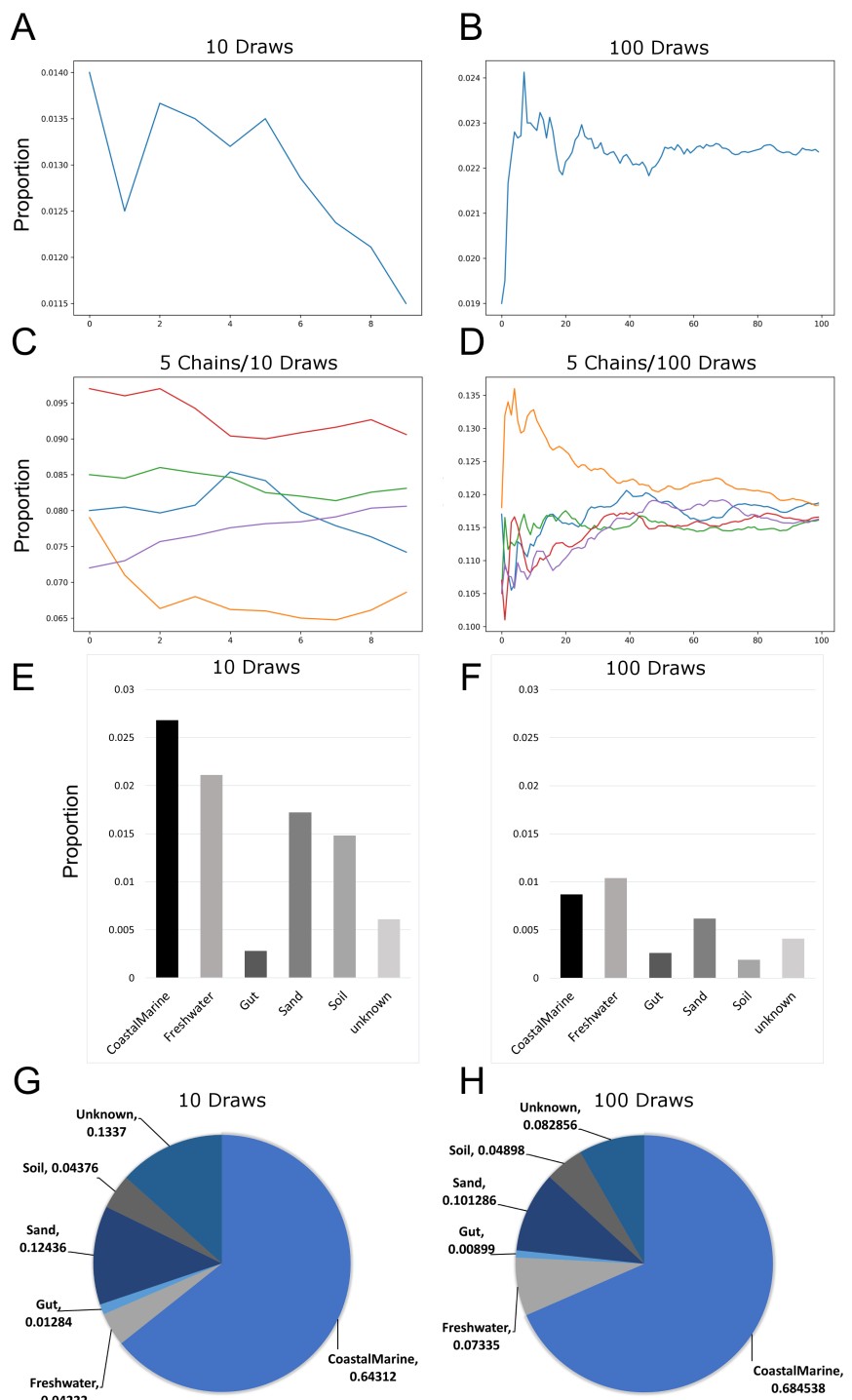

**Figure 1** **Effects of increasing Markov chain length and number on estimating source proportions for a representative coastal marine sink sample.** (A–B) Comparison between ten and one-hundred draws for a single Markov chain. The same coastal marine sample, ERR771074, was used to create both chains. (continued on next page...)

**Figure 1 (…continued)**
(C–D) Convergence of five independent Markov chains for a single sample using either ten or one-hundred draws per chain. Chains represent the estimated proportions for a given ''sink'' from a single environment or ''source''. (E–F) Absolute percent differences between the two Markov chains with the highest and lowest average proportions over all draws for each environment or ''source''. The same coastal marine sample was used with five chains and either 10 or 100 draws. (G–H) Pie charts showing proportions per source for the same single sink sample after 10 and 100 draws respectively. With 10 draws, unknown sources accounted for 13.4% of the sink sample, while after 100 draws unknown sources were 8.3% of the sink sample.

## Domain specific mSourceTracker analyses

Once we established the best general parameters for mSouceTracker, we then compared the results of combined mSourceTacker analysis to single-domain analyses of the same samples. Figure 2 shows results for a single coastal marine sink sample, ERR771074, in which mSourceTracker had been run on each specific domain (Figs. 2A–2D) and the combined dataset (Fig. 2E). The results of the bacteria alone most closely resembled the proportions from the combined metagenomic data. This is likely because the Gibbs sampling approach used to estimate the proportions would tend to pick bacterial taxa. Since the bulk of the sequences from the metagenomic data (88.4%) were bacterial. In this particular sample ''coastal marine'' comprised the largest source proportion in both the combined and bacterial fractions. A significant but lower portion of archaea and eukaryotes also came from marine environments, but the proportions were lower. In contrast, we determined an overwhelming majority of viral sequences came from a freshwater environment, while a high proportion of archaea had sand and gut origins and the eukaryotes were evenly split between the coastal marine, freshwater, sand and soil origins in this sample.

Figure 3 shows the estimated source proportions for all 14 of the coastal marine sink samples by domain. In these 14 samples, on average 50.2% of the bacteria came from the marine environments (Fig. 3A), while 42.8% of the eukaryotes had marine origins (Fig. 3D) with the remaining composition evenly distributed among the other 4 environments. Archaea samples were approximately split between the coastal marine (38.6%) and sand (30.4%) environments (Fig. 3C). Despite the fact that these samples were coastal marine, the source origins of the viruses in the metagenomes were predominantly freshwater (avg. 77.6%; Fig. 3B).

In addition to the analysis of coastal marine samples, we also tested mSourceTracker on samples collected from a built environment study of the ISS, in which the authors sampled various surfaces inside the spacecraft (*Singh et al., 2018*). mSourceTracker estimated proportions of each source environment contributing to the ISS sink samples (Fig. 4). Besides unknown sources, the bacterial taxa were predominantly from skin, averaging 24.6%, while soil contributed 2.6% (Fig. 4A). For eukaryotes, 34.3% were estimated to be from the soil environments, 5.6% from gut, and 2.9% from skin (Fig. 4B). There were too few archaea and viral taxa detected in the ISS samples to complete those domain-specific analyses.

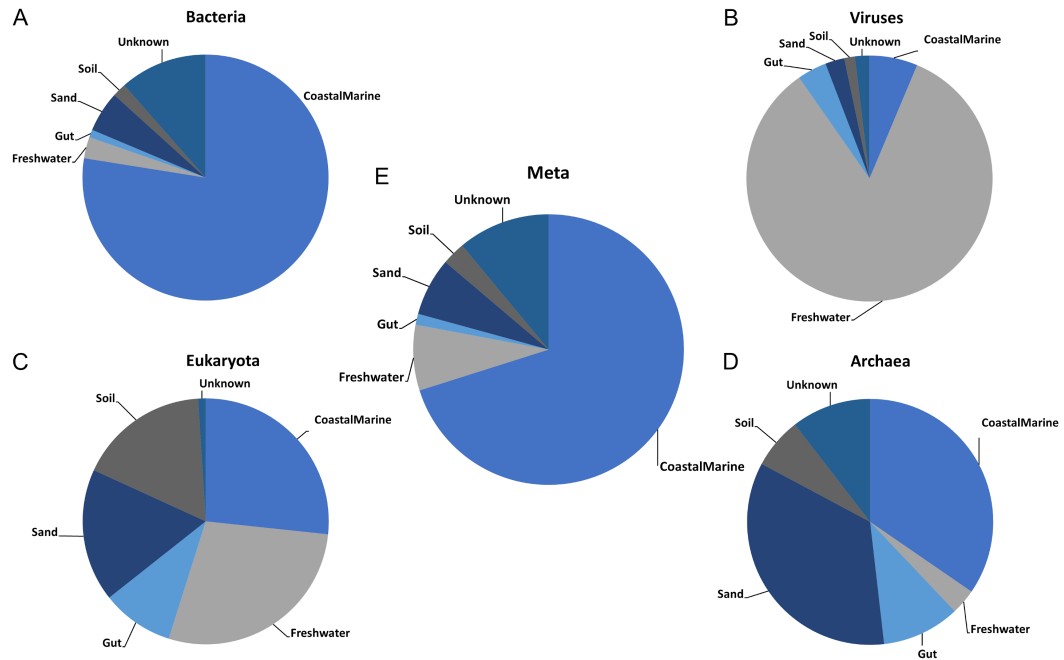

**Figure 2** **Taxon-dependent source proportion estimates in a single metagenome sample.** Graphs represent the estimated proportions from each "source" or environment for a single coastal marine "sink". Pie charts depict the estimated proportions based on the (A) Bacteria, (B) Viruses, (C) Eukaryota, and (D) Archaea in the sample (see Methods). The middle pie chart (E)"Meta" represents the estimated proportion contributed by 5 potential source environments and unknown based on the entire metagenome. The number of chains was held at 5 and the number of draws were variable so as to keep the chain differences below 5%.

## Random Forest of environments by domain

We used Random Forest to determine which organisms were best at classifying samples into each environment seen in Fig. 5. Confusion matrices for Random Forest performed on each environmental condition are shown in Fig. 6. Accuracy scores remained above 95% for all classifications and the out-of-bag error was below 5% for most samples (Table 1).

## DISCUSSION

Our results demonstrated not only the effectiveness of mSourceTracker with metagenomic datasets but also that the taxonomic diversity of metagenomic samples can potentially lend deeper insight into the mixed-source origins of complex environmental samples. The mSourceTracker analysis of the complete 14 test sink metagenomic libraries consistently revealed the biggest sources to be coastal marine, though the proportions varied considerably from sample to sample (Fig. 2). Domain-specific mSourceTracker analysis, on the other hand, often revealed patterns remarkably distinct from the combined taxa set (Figs. 2, 3). The bacterial source origins typically mirrored the full libraries (Figs. 2A, 2E), likely because the bacteria were the most abundant in all the samples. However, the other domains could be unique. For instance, mSourceTracker analysis of just the identified viruses mainly identified freshwater as the primary contributor to the

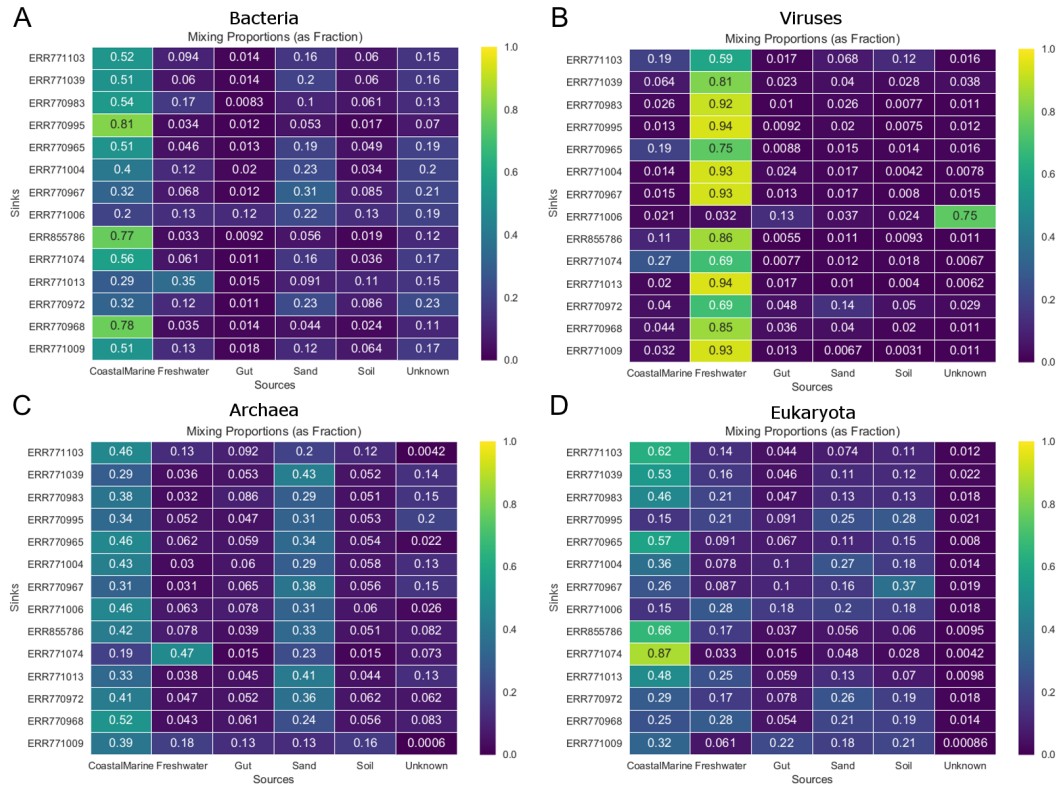

**Figure 3  Taxon-dependent source proportion estimates for 14 different coastal marine samples.** Metagenome data was separated into taxa groups (see Methods) and multiple coastal marine samples were designated as "sinks". Heatmaps produced by mSourceTracker represent the proportions of (A) Bacteria, (B) Viruses, (C) Archaea, and (D) Eukaroyta from each of the source environments. mSourceTracker default number of chains was changed to 5, and number of draws were adjusted per taxa group so absolute values between any 2 Markov chains did not exceed 5%. Five chains were used for every environment, but the number of draws was changed depending on the domain (100 draws for Archaea, 80 draws for Eukaryota, 50 draws for viruses, 20 draws for Bacteria, and 20 draws for the combined dataset).

sink diversity; according to the virus data, freshwater contributed as much as 94% of the diversity in some samples (Fig. 3B). Archaea-specific analysis typically identified both coastal marine and sand as more or less equal contributors (Fig. 2D, Fig. 3C), while the eukaryote sources were more even distributed among coastal marine, freshwater, sand, soil and even gut (22% in one sample; Fig. 2C, Fig. 3D).

The fact that domain-specific mSourceTracker analysis resulted in different source proportion estimates has two important ramifications. First, it shows that mSourceTracker can be used to identify the environmental sources of a particular group of organisms. For instance, one may conclude that, for a given sample, 75% of the viruses present originated from freshwater, while half of the bacteria were marine in origin and 28% of the eukaryotes came from soil runoff. Such results provide novel, taxon and sample-specific insight into the movement and origins of the organisms in that environment, which could be especially useful in understanding the complexity of contamination patterns or dispersal among biomes. A separate analysis of the ISS dataset found a similar pattern. In the bacterial

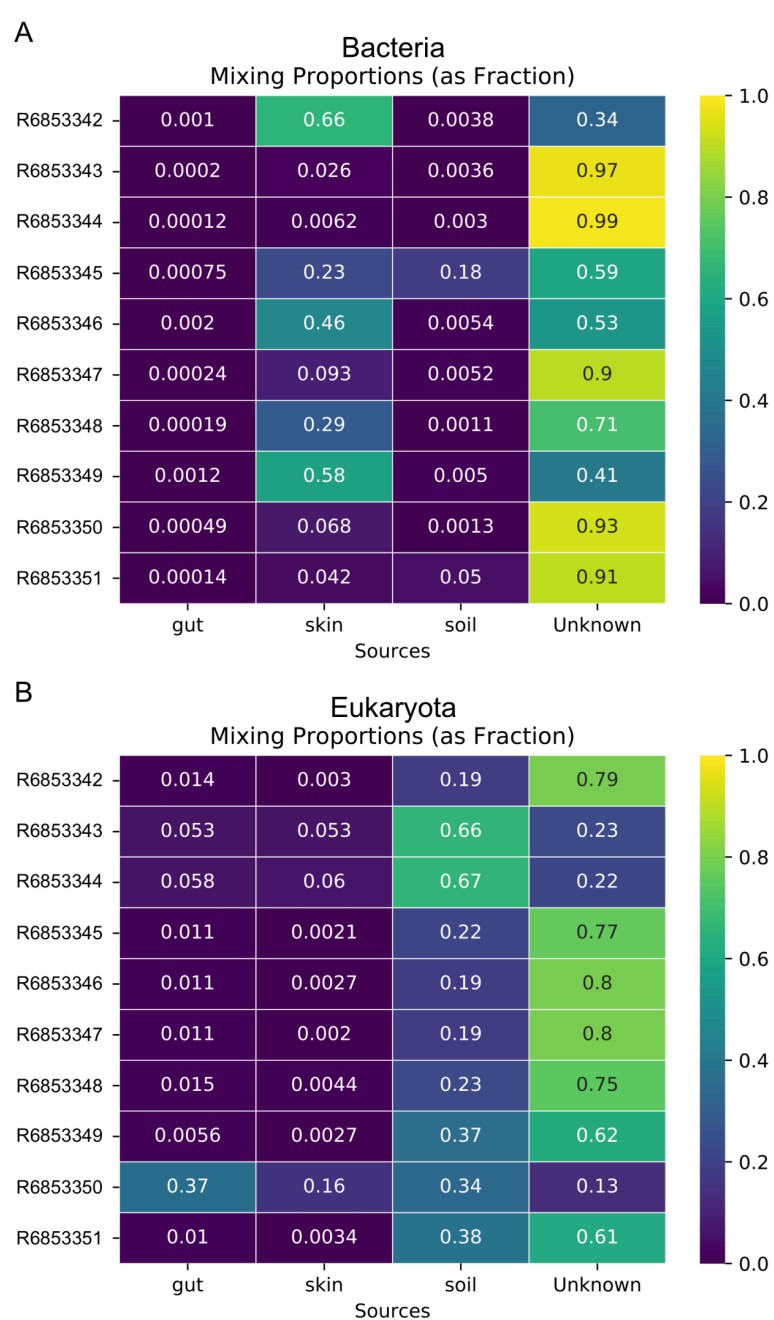

**Figure 4  Source proportion estimates for 10 different ISS samples.** Metagenome data were divided by domain (see Methods) and mSourceTracker produced heatmaps showing the proportion that each source environment contributed to each ISS sink sample for (A) Bacteria and (B) Eukaryota. Five chains and 100 draws were used in each analysis.

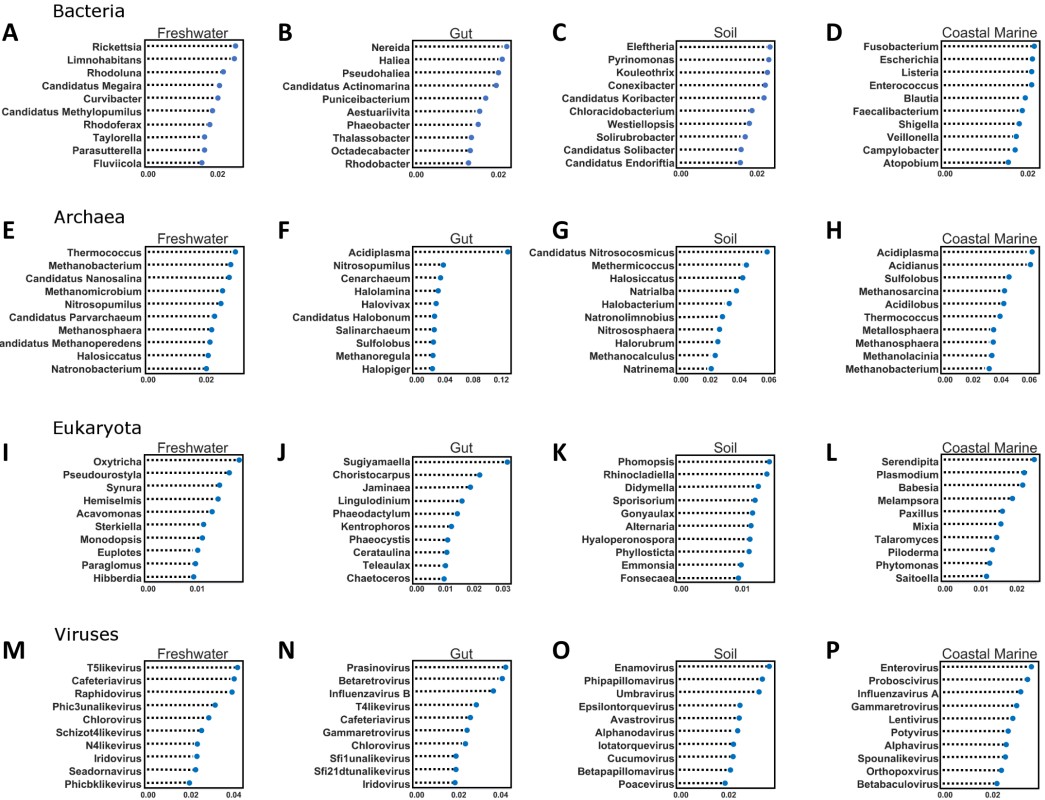

**Figure 5** **Random Forest analysis of each organismal group across 4 environments.** Random Forest analysis of each organismal group (A–D) Bacteria, (E–H) Archaea, (I–L) Eukaryota and (M–P) Viruses across four environments. Random Forest was used to determine which species were important in classifying samples as belonging to a certain environment. Each source was run against all other source classified as "other" and the data was randomly divided into testing and training subsets at approximately a 3:1 ratio. Five hundred estimators were used each time, and the 10 most important features were graphed based on relative importance.

analysis, skin was the predominant contributing source of the known environments (Fig. 4A), while soil was the main known source environment for eukaryotes (Fig. 4B). The high proportion of taxa from unknown sources was likely due to the extensive application of disinfectants used in the ISS and the clean assembly rooms, which selects for organisms from extreme environments (e.g., McMurdo Dry Valleys of Antarctica; (*Singh et al., 2018*). Inclusion of metagenomic samples from extreme environments could theoretically reduce the proportion of unknown samples.

In addition to domains, one could also easily imagine splitting datasets for mSourceTracker analysis by phylogenetic groups at a lower taxonomic level (e.g., methanogens or the proteobacteria) or even using non-organismal datasets (e.g., untargeted chemical or metabolic datasets). This is similar in principle to the approach taken by previous research studying the origins of antibiotic resistance markers (*Gou et al., 2018*; *Baral et al., 2018*; *Li, Yin & Zhang, 2018*). The identification of distinct source origins for different taxonomic groups in the same "sink" samples is not without precedent in the

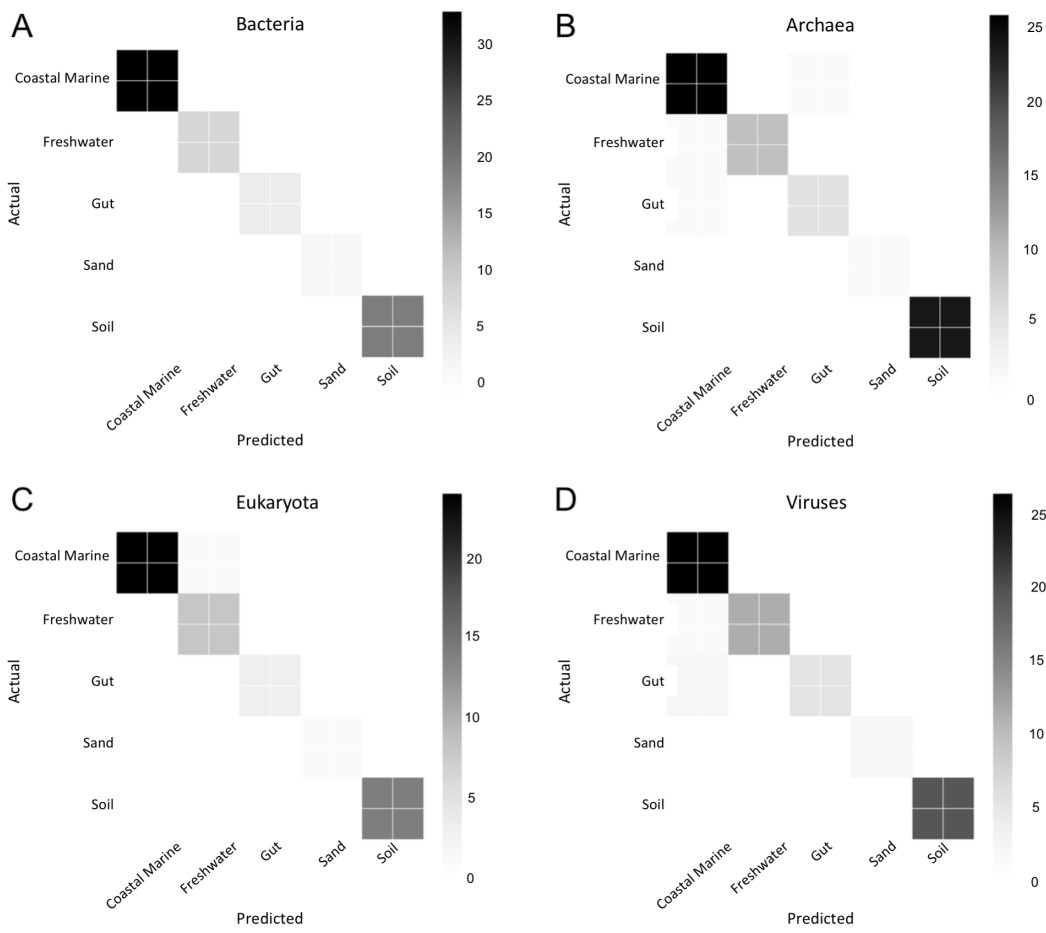

**Figure 6 Confusion matrix for organismal groups across five environments with four domains.** Heat map of confusion matrices for Random Forest analysis for each domain: (A) Bacteria, (B) Archaea, (C) Eukaryota and (D) Viruses. Data was randomly split into training and testing set at approximately a 3:1 and run using 500 estimators. Graphs display predicted source ($x$-axis) vs. the true source for ($y$-axis) for each sample in the testing dataset. The magnitude of the color indicates the number of samples tested for that condition.

literature. For example, a previous marker-gene study of restroom environments found that the fungi appeared to have radically different origins (plants and soils) than the bacteria from the same samples (human skin and gut) (*Gibbons et al., 2015*; *Fouquier, Schwartz & Kelley, 2016*). Other studies have shown very different patterns of diversity and abundance among different 'omics datasets, indicating this is a rule rather than the exception (*Bikel et al., 2015*; *Guirro et al., 2018*; *Cocolin et al., 2018*).

The second important ramification is that the diversity of environmental origins among the taxonomic domains indicates that particular taxonomic lineages may be better than others for tracking particular sources of contamination. For example, to study the input of freshwater into the coastal marine environment the viruses may be superior to the bacteria, while eukaryotes may be better for tracking soil inputs. The Random Forest analysis identified a significant number of new taxa that were highly indicative of particular

**Table 1** **Random Forest results by taxonomic group and environment type.** The columns show the accuracy and out-of-bag (OOB) error for predicting each environment type. Out-of-bag error is an estimation of the prediction error of Random Forest computed by testing each tree against data not used in building that tree.

| Domain | Environment Type | | Freshwater | Soil |
|---|---|---|---|---|
| | Gut | Coastal Marine | | |
| **Bacteria** | | | | |
| Accuracy | 0.982 | 0.984 | 0.97 | 0.985 |
| OOB error | 0.022 | 0.026 | 0.052 | 0.0174 |
| **Archaea** | | | | |
| Accuracy | 0.966 | 0.977 | 0.966 | 0.98 |
| OOB error | 0.0497 | 0.062 | 0.049 | 0.021 |
| **Eukaryota** | | | | |
| Accuracy | 0.955 | 0.967 | 0.959 | 0.984 |
| OOB error | 0.047 | 0.062 | 0.066 | 0.034 |
| **Viruses** | | | | |
| Accuracy | 0.964 | 0.927 | 0.96 | 0.965 |
| OOB error | 0.054 | 0.061 | 0.058 | 0.027 |

environments (Fig. 5). For every domain in every environment, we were able to identify certain features (taxa) that contributed significantly to the classification of the environment. In the future, such taxa could be used singly or in combination to detect particular types of contamination. This is the same principle used by culture-based source-tracking that tracks fecal contamination using strains of *E. coli* (*Ravaliya et al., 2014*). Recently, *Stachler & Bibby (2014)* proposed using sequences of crASSphage as a highly specific indicator of human fecal contamination (*Stachler & Bibby, 2014*).

One important caveat of mSourceTracker method is the general challenge of identifying taxa from metagenomic datasets. It is well known that much of the sequences from metagenomic datasets are not currently identifiable because databases are incomplete. Unlike 16S, it is not possible to put all the sequences from a library into a phylogenetic context, so many of them remain unknown and not currently useful in mSourceTracker analysis. As databases grow, this problem should diminish. The other issue is one of identification itself. There are many methods of identifying reads from metagenomic libraries, both alignment and k-mer based, and sometimes they can give very different results for the same samples (*Quince et al., 2017*). We expect that this may have a profound effect in some cases, and future research should look at the importance of identification algorithms and databases on mSourceTracker results. Finally, in order for mSourceTracker to be broadly applicable, it will be critical to have many more metagenome datasets collected from specific environments. Large environmental collections such as the Earth Microbiome Project make it easy to find 16S ribosomal RNA libraries for any given environment, and it is relatively cheap to create many libraries in any given study and the analysis is easy to perform on a laptop. As the costs of sequencing continues to decline and the computational power and number of available datasets increases, the mSourceTracker approach will become increasingly tractable and commonplace.

## CONCLUSIONS

In this study, we demonstrated three findings: (1) mSourceTracker is a straightforward and effective application of SourceTracker for determining source proportions using shotgun metagenomic datasets; (2) our chain convergence tests and visualizations allow researcher to identify when estimates do not converge, which mainly occurred when source datasets had poor taxonomic coverage; and (3) the purposeful domain-specific subdivision of metagenomic datasets has the potential to lend powerful new biological insights into the source and movement of microorganisms among environments. While our analyses demonstrated SourceTracker's utility and potential with metagenomic data, the results are only as good as the input data allow (the ''garbage in, garbage out'' rule). All inferences based on metagenomics data are dependent on the extent and quality of existing databases and the effectiveness on taxonomic identification approaches. Methods other than Kaiju and more extensive databases could certainly produce different results and hopefully reduce the proportion of unknowns in the estimates. We also note that some of our source sample sets of metagenomes were small; increasing the sample size, purity and number of source datasets could also have a significant impact on interpretations. Investigation of all these parameters is beyond the scope of this study, which is focused on mSourceTracker development and proof of principle. However, such factors should be taken into account in future studies.

## ACKNOWLEDGEMENTS

We thank E. Dinsdale for helpful insights and comments on the manuscript. We also thank P Torres for guidance with the original SourceTracker, B Ho for help with cluster computing and R Edwards for allowing us an account on the anthill computer cluster at San Diego State University.

### Funding

This work was funded by a fellowship awarded to Scott T. Kelley by the Alexander von Humboldt Foundation of Germany. The funders had no role in study design, data collection and analysis, decision to publish, or preparation of the manuscript.

### Grant Disclosures

The following grant information was disclosed by the authors:
Alexander von Humboldt Foundation of Germany..

### Competing Interests

The authors declare there are no competing interests.

### Author Contributions

- Jordan J. McGhee and Laura Sisk-Hackworth performed the experiments, analyzed the data, prepared figures and/or tables, authored or reviewed drafts of the paper, and approved the final draft.

- Nick Rawson performed the experiments, analyzed the data, authored or reviewed drafts of the paper, and approved the final draft.
- Barbara A. Bailey and Antonio Fernandez-Guerra conceived and designed the experiments, authored or reviewed drafts of the paper, and approved the final draft.
- Scott T. Kelley conceived and designed the experiments, prepared figures and/or tables, authored or reviewed drafts of the paper, and approved the final draft.

## Data Availability

Data and code are available at GitHub: https://github.com/residentjordan/SourceTracker2-diagnostics.

## Supplemental Information

Supplemental information for this article can be found online at http://dx.doi.org/10.7717/peerj.8783#supplemental-information.

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
