# Peer review of "Meta-SourceTracker: application of Bayesian source tracking to shotgun metagenomics"

_PeerJ, doi:10.7717/peerj.8783_

## Round 0.1 · original submission · Major Revisions

Before we can make a final decision on the acceptance of your manuscript for publication, a number of major revisions are necessary as described by the three reviewers. In particular, please be sure to discuss the differences between Meta-SourceTracker and SourceTracker, emphasizing the new features provided in Meta-SourceTracker.

Reviewer 1 ·

Basic reporting

The language could be edited to be more understandable by non-experts and those unfamiliar with using SourceTracker.

Experimental design

SourceTracker is highly dependent on the dataset used for training – the study design here is lacking in adequate number of samples across the different source and sink environments for evaluating the algorithm’s performance.

One of the main advantages of using shotgun metagenomics over 16S rRNA gene sequencing is identification of organisms at the species level - however, this manuscript primarily used Kingdom-level and Genus-level information.

The authors used 1000 sequences per sample – this is the sequencing depth used in the Sourcetracker tutorial, but is not appropriate for use with shotgun metagenomics (or necessarily real 16S rRNA gene sequencing datasets). In shotgun metagenomics, the number of reads that are required for accurately identifying organisms is much higher since the data is more complex.

There is very little detail regarding the processing of the metagenomics data prior to use with SourceTracker.

Validity of the findings

More robust evaluation of SourceTracker parameters (sequencing depth, draws, etc.) would strengthen this work. 10 and 100 draws were compared, but more draws may lead to more convergence. The sampling depth also will likely alter the stability of the SourceTracker results.

Additional comments

This manuscript does not seem to extend the existing SourceTracker program, which was expected based on the name, abstract, and introduction. A benchmarking study on using SourceTracker with shotgun metagenomics may be a useful contribution to the field, but this should not be presented as a new or development of a software package. Further, more parameterization evaluation should be performed for a benchmarking study.

·

Basic reporting

McGhee et al. in their paper “META-SOURCETRACKER: APPLICATION OF BAYESIAN SOURCE TRACKING TO SHOTGUN METAGENOMICS” present an original method to track the origin of contaminating bacteria and other domains, separately or in all together, using shotgun metagenomic datasets.
They used mostly clear, unambiguous, professional English language. Some terms need more clarification and they are listed below. The Intro and background describe well the context with literature well referenced.

Major modifications:
• They need to describe the word “sink” and “sources” to the readers (L148-00149) as well “draws” and “chains” (L152). They could change the order between the two sections (“Simulation data for mSourceTracker analysis” and mSourceTracker: diagnostic add-on feature) if it help them to present these words.
• Which Gibbs sampling data are they talking in L158? It is from SourceTracker or mSourceTracker?
• List of 14 samples among the 110 downloaded coast marine data used for simulation data (L147 and L196)
• L202 do they talk about mSourceTracker or SourceTracker (Knights et al 2011)?
• Do the 14 sources are 14 sinks that they used in the simulation?
• They need to provide which single coastal marine sink they used in L233.
• Replace the color in confusion matrix by the values or combine both of them. They need to list the samples used in each environment as “sink” and as “sources”
• Figure 1 need further adjustment to help the readers to understand it. Need to provide the name of coastal marine sample used in Fig 1. A, B, D and “source” used too.
• Fig 1. C should be separated to Fig 1. A, B, D as it is done on more datasets and the choice of datasets for this figure need to be described in the result section and listed in the legend as well.
• Fig. 1 D need more description in the result section to help the readers to interpret this figure and values provided.

Minor modifications:
• Remove “_” line 60
• Move paper “Knights et al, 2011) L80-81 to the end of the first sentence L75
• Figure 3. Do they talk SourceTracker or mSourceTracker?
• Legend of Table 1 put “(OOB)” after “out-of-bag error” and add the meaning of this.
• List the 4 environments in the legend of Figure 5.

Experimental design

This paper is an original primary research within scope of the journal “PeerJ”.
The research question is well defined and stated how they fill an identified knowledge gap.
The Method and data are partially described and need additional information before any publication:
1) the authors should provide a descriptive about their software and the Bayesian method as they put in the title as they never describe the method implemented in their software
2) the authors should provide a link to for webpage or github link that should contain:
a. bin script of mSourceTracker as well as the programming script talking L122
b. user manual with
i. examples how to run their software and programming script with data presenting in this manuscript to allow users to reproduce their findings (related to “Simulation data for mSourceTracker analysis” and mSourceTracker:diagnostic add-on feature) and then figures
ii. format of input data to provide to run the software and output data
iii. how to create and update a dictionary (related to “Taxonomic separation of metagenomics”)
c. installation manual
3) The authors should provide more information about storage, memory usage, computing usage, and wall-time consuming to help the users to check and use their software in the best conditions. They can provide these pieces of information based on their data used in their manuscript.
4) They need to provide the parameter values used in their analysis and how they make this choose (L 223-225)
5) They need to provide the parameter values used (L231)
6) They put in the script “gibbs.py” a comment “Gibb's sampler for Bayesian estimation of microbial sample sources. For details, see the project README file.” However, we don’t have access to such README file

Validity of the findings

Although the data are described in the Supplementary Table 1, it needs further information.
1) add information what they are these accession number and where we can download them
2) add some meta-data such as geographic location of data or put an earth map with dot where data come from

The data appear to be robust, statistically sound, and controlled, but better description of their method will help to confirm it. Conclusions are well stated, linked to original research question & limited to supporting results

Additional comments

This paper presents a method for tracking contaminations using metagenomic data. Before publication, further pieces of information are needed to add in order to allow users to use it and reproduce their findings.

·

Basic reporting

The authors did a good job to explain background and need to use metagenomic data for microbial source tracking.

Overall the manuscript is well written. All the figures and tables are easy to read.

Citation of SourceTracker should be added in line 75 after its first introduction, instead of line 80.

I would suggest that the authors also discuss 1) if there are other softwares that can be used for microbial source tracking using metagenomic data and 2) if so, what are the pros and cons of existing softwares and the unique features of mSourceTracker.

There are some formatting and language issues that should be addressed.

Line 60, "_ _" before "ecosystem function" should be deleted.
Line 77, a comma should be added before "which". Actually, please check the usage of "which" throughout the whole manuscript.
Line 93, change "allows" to "allow"
Line 94, change "eukaryotes" to "eukaryota"
Line 115, change the comma after "this study" to something like "this study, which includes"
Line 150, change "which" to "that" after "Samples"

Experimental design

Enough samples were included in the analysis.

Different types of samples were included.

Domain-specific analysis is especially interesting and provided interesting results.

The selection of 10 draws and 100 draws clearly showed the impact of draws on proportion estimates. Please justify why these two numbers were selected and potential issues of using high number of draws. Will calculation time be significantly increased with 100 draws? How about 200 draws? Ideally, three number of draws can be tested and results can be compared to identified the optimal number of draws.

Validity of the findings

I like the domain-specific results very much. The authors did a good job to show how the mSourceTracker can be used to analyze domain-specific information. When a microbial risk assessment is conducted, the domain-specific information will be very useful.

Is it possible to identify group-specific, family-specific, or even species-specific information from mSourceTracker? It would be a nice feature.

The limitation of current study is well explained. I agree with the authors on the challenges of using limited databases and algorithms. However, the "pure" environmental databases will be impossible to collect because environment is an open system and will always be affected from all different types of samples. Please further explain the meaning of "pure" or elaborate the idea.

Additional comments

Overall this is a well written manuscript. The authors expanded their existing work on the popular software SourceTracker to analyze metagenomic datasets. The results could provide interesting insights on microbial source tracking.

---

## Round 0.2 · Minor Revisions

Please carefully review and respond to the critiques of Reviewers 1 and 2 below. As indicated by the response of Reviewer 1, there is still confusion regarding the differentiation between SourceTracker2 and Meta-SourceTracker. In fact, the program SourceTracker2 is not mentioned at all other than in the name of the GitHub repository. (Will there be a separate publication describing SourceTracker2? If so, it might be worth indicating that this will be forthcoming.) One possible approach would be to describe Meta-SourceTracker as a novel application of SourceTracker rather than an "expansion" of SourceTracker.

Reviewer 1 ·

Basic reporting

The authors were responsive to all the reviewers’ comments and have made edits to address concerns that were raised. Despite this, the manuscript can use improvement.

The language is still very technical and could be made more accessible.

The contribution of the presented work should be more accurately stated - evaluating/benchmarking Sourcetracker for use with shotgun metagenomic data, not expanding or building on Sourcetracker. Referring to the analysis performed as meta-Sourcetracker or mSourcetracker makes it sound as though new software was created, when it does not appear that this is the case based on the sourcetracker code presented in the corresponding github repo, which is identical to the sourcetracker code on Dan Knights' lab github.

The code to reproduce the presented analyses should be made available to readers wishing to attempt this on their own (the corresponding github lacks this key component) with metagenomic data.

From the Github page, it is not clear what is from the original Sourcetracker and what the contribution is. The Github page needs more development such as the scripts, sample test data, an instruction manual, and supplementary analysis results referred to on line 207.

Experimental design

The addition of diagnostic plots is nice, but the plots need axis labels and the reader/ user should be given some orientation as to how to interpret the plots.

Validity of the findings

In the rebuttal, it is stated that a more comprehensive evaluation of parameters was performed, but these results are not presented.

The differences between Sourcetracker run on each type of kingdom independently and on the overall metagenomic data are likely different due to sampling depths, to actually compare both of these approaches, the test tests would have to have 1,000 species from each kingdom or 4,000 species across all kingdoms.

·

Basic reporting

They answer to the different comments mentioned to them. Although they increase the size of image to improve the reading, the image after printing are blurry and it is difficult to read correct the text. The authors need to improve the resolution of image to avoid this issue for the published version.

Please in the GitHub README, replace "meta data" by "metagenomic data"

Experimental design

1) in the comment of the function "ConditionalProbability", they put "The formula used to calculate the conditional joint probability is described in the project readme file." However, no such file available in the file "sourcetracker.py". Could they clarify where this file could be found?

2) Line 165, the authors should link to an example of output table that mSourceTracker need. Moreover, they should provide in the name of their programming script that allow to produce this output file in Line 143 and put an example how to use this script in their GitHub

3) The author need to remind the readers in the beginning of the section "mSourceTracker: diagnostic add-on feature" that mSourceTracker is a new option of sourceTracker2 called "--diagnostics" where data, script, instruction are available in their GitHub (as they put in their last sentence of this section). Moreover, they need to put in the beginning of this section a sentence to explain the aim of this option.

4) It needs to be clarify also in the section "mSourceTracker: diagnostic add-on feature" if the option "--diagnostics" is available in the original GitHub folder of sourceTracker2 (https://github.com/biota/sourcetracker2) and if the other options of sourceTracker2 could already deal metagenomic data.

5) It will be great if they could provide an template script for at least one dataset how they perform their random forest analysis and confusion matrices to help in the reproduction of their findings.

Validity of the findings

no comment

Additional comments

The authors has overall well written the manuscript to explain the new option they developed on the software SourceTracker to analyze metagenomic dataset and provide two examples how this new option could provide an interesting insights on microbial source tracking based on the reference metagenomic datasets proposed to be the sources. This highlights also the needs to work to improve the characterization of unknown reads. In this idea, could msourceTracker annotate which source the reads of sink samples belong to and then allow extracting, for example, only the unknown reads to work on this sub-dataset? If this option is already available, please highlight it in your paper.

·

Basic reporting

Overall the study is well written and the authors' responses are satisfactory.

Experimental design

The number of draws is important and maybe the default value can be changed to 100 based on the current study.

Validity of the findings

The revised statement is accurate and reflected the challenges of environmental samples.

Additional comments

The authors have addressed my comments.

---

## Round 0.3 · accepted · Accept

Thank-you for your revised manuscript. Your revisions, based on reviewer comments, have improved the manuscript and helped to clear up some of the previous confusion regarding how the work reported here can be differentiated from past publications describing SourceTraker.